# Severe Intra- and Post-Operative Lactic Acidosis in a Patient Who Underwent Robotic Thoracoscopic Surgery

**DOI:** 10.3390/biomedicines13030568

**Published:** 2025-02-24

**Authors:** Alexander Smirnov, Michael Semionov, Shlomo Yaron Ishay, Alexander Zlotnik, Vadim E. Fraifeld, Dmitry Frank

**Affiliations:** 1Department of Anesthesiology, Soroka University Medical Center, Faculty of Health Sciences, Ben-Gurion University of the Negev, Beer-Sheva 8410101, Israel; smirnovalexander69@gmail.com (A.S.); semyonov.michael@gmail.com (M.S.); alexander.zlotnik.71@gmail.com (A.Z.); 2Department of Cardiothoracic Surgery, Soroka University Medical Center, Faculty of Health Sciences, Ben-Gurion University of the Negev, Beer-Sheva 8410501, Israel; yaronsi@clalit.org.il; 3The Shraga Segal Department of Microbiology, Immunology, and Genetics, Faculty of Health Sciences, Center for Multidisciplinary Research on Aging, Ben-Gurion University of the Negev, Beer-Sheva 8410501, Israel; vadim.fraifeld@gmail.com

**Keywords:** severe lactic acidosis, robotic thoracic surgery, thiamine deficiency, chemotherapy, Warburg effect

## Abstract

**Background/Objectives:** Lactic acidosis is one of the most common causes of metabolic acidosis in hospitalized patients. It happens when lactic acid production exceeds lactic acid clearance. The elevation of lactate was commonly improved after the restoration of tissue perfusion. However, there are rare cases of severe lactate elevation (greater than 8 mmol/L) in the intraoperative period of thoracoscopic surgery. A poor prognosis with high morbidity and mortality characterizes these cases. **Case Description:** A 72-year-old man was admitted to the Soroka University Medical Center for thoracoscopic robotic left upper lobe lobectomy due to squamous cell carcinoma. At the end of surgery (overall, 8.5 h), the lactate level reached 10.2 mmol/L with the development of severe lactic metabolic acidosis. Thiamine was successfully given to patients to stimulate lactate clearance towards the cycle of tricarboxylic acids via pyruvate. **Conclusions:** Though the pathogenesis of this state in our case is not fully clear, it may have been induced by chemotherapy and during tumor manipulation by a surgeon. The successful recovery of blood lactic levels after thiamine treatment is suggestive of thiamine deficiency as a possible cause of lactic acidosis in our patient. Although we do not have data on the plasma thiamine level, we suggest that its determination in the perioperative period would be beneficial for excluding a probable thiamine deficiency in the case of severe lactic acidosis.

## 1. Introduction

Lactic acidosis is one of the most common causes of metabolic acidosis in hospitalized patients. Intraoperatively, lactic acid accumulation in the blood can cause a high anion gap metabolic acidosis and precipitate negative hemodynamic and metabolic consequences [1]. It happens when lactic acid production exceeds lactic acid clearance. The increase in lactate production can result from many conditions, such as impaired tissue oxygenation, decreased oxygen delivery, or a leak in mitochondrial oxygen utilization [1,2]. It should be emphasized that lactic acid plays an essential role in energy metabolism, being involved in the tricarboxylic acid cycle. L-lactate is a dominant isomer in mammals, which is primarily oxidized to carbon dioxide and water (70 to 80 percent) and used to generate glucose (15 to 20 percent). Utilization of lactic acid mainly occurs in the liver, but it also appears in the kidneys and the heart [2].

Lactic acidosis was described in numerous intraoperative cases [3]. In most cases, the lactate elevation was mild to moderate (less than 8 mmol/L) and occurred as a consequence of tissue hypoperfusion (type A lactic acidosis). This elevation was commonly improved after the restoration of tissue perfusion. However, there are rare cases of severe lactate elevation (greater than 8 mmol/L) in the intraoperative period of thoracoscopic surgery. A poor prognosis with high morbidity and mortality characterizes these cases [1,4]. Several operation types can induce lactic acidosis with transient elevation of lactate, such as surgeries using cardiopulmonary bypass like CABG and aortic surgeries, major hepatectomy, and orthopedic surgery with a tourniquet [5,6]. There are no reports of severe metabolic acidosis in robotic or non-robotic thoracic surgeries.

Here, we report for the first time about severe intraoperative lactic acidosis unexpectedly developed in the intra-operative period in a patient under thoracoscopic surgery.

## 2. Case Description

A 72-year-old man was admitted to the Soroka University Medical Center for thoracoscopic robotic left upper lobe lobectomy due to squamous cell carcinoma PDL < 50%, which had been diagnosed six months previously after a biopsy.

Medical history: ischemic heart disease, status post stent implantation to LAD in 2002. Transthoracic echocardiography demonstrated a good LV systolic function with no evidence of significant valvular pathology. Currently, sinus rhythm after cardiac ablation because of paroxysmal atrial fibrillation; arterial hypertension, hyperlipidemia, diabetes mellitus type II, former smoker 40 PY, moderate COPD (FEV1—73%, FEV1/FVC 65, DLCO 82%). Current medical treatment—Eliquis, Rosuvastatin, Jardiance, Alfuzosin, Duplex, Nexium, Normalol.

He completed three courses of neoadjuvant chemotherapy with Nivolumab, Paclitaxel, and Carboplatin and immunotherapy with Pembrolizumab before surgery with an excellent clinical and radiological response. The patient was advised to stop treatment with Eliquis and Jardiance 48 h and 72 h before surgery.

The day before surgery, the patient was admitted to the Thoracic Surgery Department and examined by an anesthetist. There were no clinical signs of dyspnea, elevated temperature, or other abnormal findings. Lab data: Hb 12.6, INR 1.01, creatinine 1.2, glucose 117, and normal hepatic function. ECG showed a normal sinus rhythm, 80 bpm, with no signs of myocardial ischemia.

In the operating room, after induction of general anesthesia with Propofol, Fentanyl, and Rocuronium, a double-lumen tracheal tube N39 French (Medtronic, Minneapolis, MN, USA) was used for lung separation. An arterial line and two peripheral vein catheters were inserted, and a urinary catheter was also applied. In addition, thoracic epidural anesthesia was performed at T7 for post-operation multimodal analgesia. The initial ABGs were pH 7.4, PO_2_ 150 mm Hg, PCO_2_ 40 mm Hg, HCO_3_ 27 mEq/L, and lactate 1.2 mmol/L.

The first two hours of the surgery passed without complications; the urine output was below 0.5 mL/kg/h, and the overall fluid balance was restricted to 100 mL/h per 70 kg of body mass to prevent down-lung syndrome, lung edema, and enhancement of gut recovery, according to ERAS guidelines for thoracic surgery [7]. The end-tidal CO_2_ varied within the normal range (35–45 mm Hg) in the perioperative period. The ventilation parameters, such as tidal volume (TV) and peak inspiratory pressure (PIP), were also accepted for one-lung ventilation protocol (TV 5–6 mL/kg and PIP ≤ 30 mm H_2_O). The patient was hemodynamically stable, with minimal phenylephrine support throughout the perioperative period. Blood pressure variations were within 15–20% of its baseline, and mean arterial pressure was >65 mm Hg. If needed, the transient decrease in the patient’s blood pressure was corrected by administrating small boluses of phenylephrine at 50–100 mcg. The total dose was quite small and did not exceed 300 mcg. The pulse pressure variation was normal and did not exceed 13% throughout the surgery. After two hours, the blood level of lactate unexpectedly rose to 2.5 mmol/L, progressively elevated, and peaked at the end of the surgery. At the end of surgery (overall, 8.5 h), the lactate level reached 10.2 mmol/L with the development of severe lactic metabolic acidosis. Because of health-threatening conditions, fluid boluses (in total, 2 L) were given, with no improvement in lactate levels and elevation in urinary output.

Because of the patient’s position and restricted fluid balance, a moderate elevation in blood lactate level is not unusual. Yet, in our case, it was accompanied by oliguria and progressive metabolic acidosis with decreased bicarbonate levels (Figure 1). To stimulate kidney function, we initially administered a 500 mL fluid bolus, but without any visible response; the patient remained oliguric and acidic.

When lactate levels reached 5 mmol/L, we continued liberal fluid therapy. Since the glucose level only mildly rose (to 160 mg/dL), the insulin treatment was postponed. Administration of Furosemide (20 mg IV) did not result in an increase in urine output. Because of restricted access to the patient during the surgery, cardiac ultrasound examination was delayed until the end of surgery.

After 2.5 L of fluid therapy, the patient remained oliguric with progressively elevated lactate levels and worsening metabolic acidosis. At the end of the surgery, after tumor resection, the ABGs were as follows: PH 7.2, PCO_2_ 38 mmHg, PO_2_ 210 mmHg, HCO^−^ 10.5 mEq/L, Hb 11.5 g/dl, K 4.9 mEq/L, Na 138 mEq/L, and lactate 11.2 mmol/L. The core temperature was 36 °C, with no response to treatment. Urine analysis for ketone bodies was slightly positive. During the surgery, the patient was hemodynamically stable. Metabolic relationships through the case between pH, HCO_3_^−^ and lactate are represented in Figure 2.

After discussing the situation with the surgeon and ICU provider, we transferred the patient to the Recovery Room and then to the ICU for further diagnosis and treatment. In the ICU, he was sedated, ventilated, and received fluid treatment. A complete blood panel showed normal hepatic function and a slightly increased creatinine level (1.6 mg/dL with 1 mg/dL—base). The ECG showed no signs of ischemia. Troponin levels were normal. The patient was hypothermic, with a body temperature of 35.5 °C. In the intraoperative period, the positive pressure of 5–8 mm Hg in the thoracic cavity was established by insufflation of Carbon dioxide (CO_2_).

Thiamin is an essential co-factor for pyruvate conversion and, when deficient, can aggravate lactic acidosis [8]. Thiamine was successfully given to patients with lactic acidosis [9] to stimulate lactate clearance towards the cycle of tricarboxylic acids via pyruvate (Figure 3). With this in mind, we initiated the thiamine treatment at 100 mg daily. Within one hour of treatment initiation, the patient’s urine output increased to 200 mL/h, and lactate levels gradually decreased to the normal range. A liberal fluid treatment was continued, and the ABG normalized. The patient was successfully extubated 24 h after admission to the ICU and transferred to the surgical unit the day after.

## 3. Discussion

The most common causes of lactic acidosis are associated with tissue hypoperfusion [10] because of hypovolemia, cardiac failure, sepsis, cardiopulmonary arrest, and thromboembolic events. However, the patient was hemodynamically relatively stable during surgery. Paroxysmal atrial fibrillation in the patient’s history may lead to thrombus generation and thus should be taken into consideration for differential diagnosis. This possibility, in particular a mesenteric event, was ruled out because of negative clinical findings and improved lactate levels in the ICU, so that CT angiography was not relevant in our case. Also, septic shock was not relevant in our case due to the elective surgery.

Another frequent cause for lactic acidosis could be related to impaired lactate clearance, including hepatic dysfunction, hypothermia, drug-induced lactic acidosis, and thiamine deficiency [11]. There were no signs of abnormal hepatic function. Hypothermia was transient and apparently could not cause severe lactic acidosis. Among the drugs that could promote acidosis, our patient permanently received Jardiance. However, this drug could induce ketoacidosis but not lactic acidosis [12]. In our case, the drug was stopped 72 h before surgery, and only trace urine ketones were detected. Thus, Jardiance at the dose is unlikely to result in severe metabolic disturbances. The improvement of the acid–base balance after thiamine treatment could be considered therapy “ex juvantibus”, indicating a possible role of thiamine deficiency in developing severe lactic acidosis in our case. However, we observed no overt signs of the thiamine deficiency’s neurological manifestations. Moreover, the improvement in lactate concentration was started in the post-operative care unit (PACU) several hours before initiation of thiamine treatment. Yet, due to the successful recovery of blood lactic levels after thiamine treatment, it could be speculated that thiamine deficiency might be a possible cause of lactic acidosis in our patient. Although we do not have data on the plasma thiamine level, we suggest that its determination in the perioperative period should also be considered in similar cases. Unfortunately, there are no laboratory data on thiamine levels in our case.

Single-lung ventilation can potentially contribute to transient regional hypoxia and ischemia in the non-ventilated lung, with a subsequent increase in lactate levels. Yet, based on our experience of approximately three hundred thoracic surgeries per year and the corresponding protocols on one-lung ventilation (OLV), such a dramatic increase in lactate levels is hardly probable. The pulmonary vasoconstriction mechanism (PVC) reduces the lung blood supply to a non-ventilated lung, thus partially adjusting O_2_ delivery to tissue demand. All in all, this commonly does not result in lung ischemia and lactic acid accumulation secondary to PVC [13].

A point for further consideration is lactic acidosis, which rarely occurs in patients with leukemia, lymphoma, and solid epithelial tumors [14,15]. Of note, there are only two case reports regarding refractory lactic acidosis in small-cell carcinoma of the lung without surgical manipulation [16,17]. It is attractive to speculate that lactic acidosis may be associated with the so-called “Warburg effect”. The matter is that, in contrast to many normal cells, neoplastic cells primarily rely on anaerobic glycolysis, generating increased lactate levels. The lactate release could be enhanced during tumor manipulation by a surgeon. In our case, it could, to some extent, contribute to lactic acidosis, since it appeared at the time of tumor resection. In addition, standard chemotherapy per se could induce a metabolic shift towards anaerobic glycolysis. The programmed cell death protein 1 (PD-1) inhibitors (Nivolumab, Pembrolizumab) may promote lactate accumulation (like the standard chemotherapy) by inhibiting pyruvate dehydrogenase in mitochondria [18]. Also, the PD-1 inhibitors could induce subclinical thiamine deficiency with a subsequent increase in lactate levels (Figure 3). In particular, in one lung cancer study, thiamine deficiency was reported when the patient received Nivolumab [19]. Altogether, preoperative screening could be helpful in patients with prior chemotherapy exposure to PD-1 inhibitors.

## 4. Conclusions

We describe for the first time a case of severe lactic acidosis developed at the time of thoracoscopic robotic lung lobectomy because of squamous cell carcinoma. Though the pathogenesis of this state in our case is not fully clear, it may have been induced by chemotherapy and during tumor manipulation by a surgeon. The successful recovery of blood lactic levels after thiamine treatment might point towards thiamine deficiency as a possible cause of lactic acidosis in our patient. Although we do not have data on plasma thiamine levels, we suggest that determining them in the perioperative period could be beneficial for robotic thoracoscopic surgeries. Whatever the case, thiamine prophylaxis in chemotherapy-exposed patients who are planned for robotic thoracoscopic surgeries warrants further investigation.

## Figures and Tables

**Figure 1 biomedicines-13-00568-f001:**
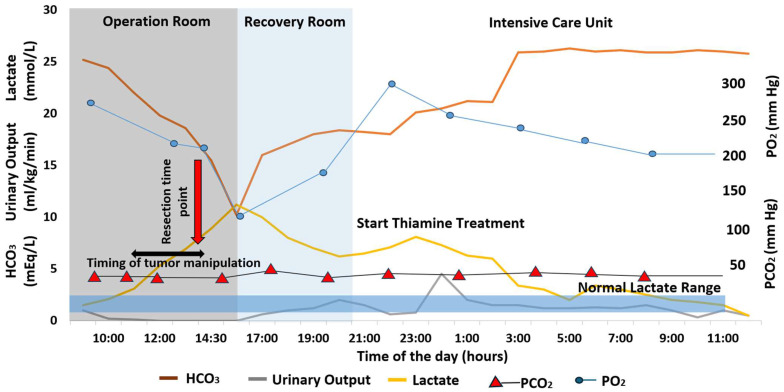
Metabolic changes in intra- and post-operative periods.

**Figure 2 biomedicines-13-00568-f002:**
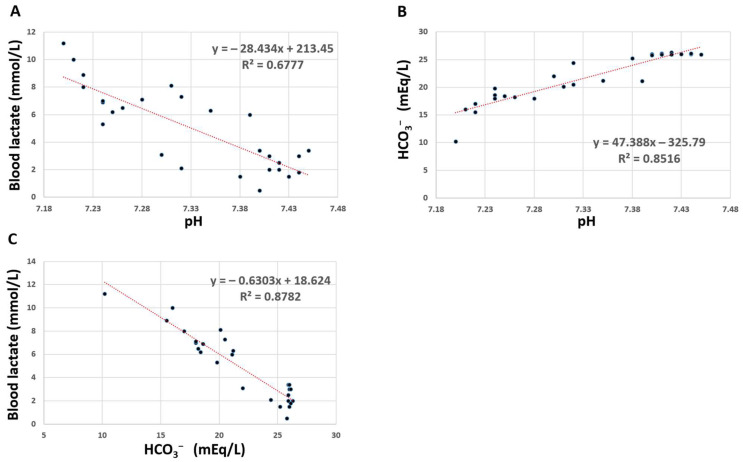
Metabolic relationships between pH, HCO_3_^−^, and lactate in intra- and post-operative periods of the patient. (**A**) Lactate vs. pH. (**B**) Bicarbonate vs. pH. (**C**) Lactate vs. bicarbonate. Each dot represents the measured parameter at axis Y vs. the corresponding variable at axis X. The red lines are a graphical representation of linear regression.

**Figure 3 biomedicines-13-00568-f003:**
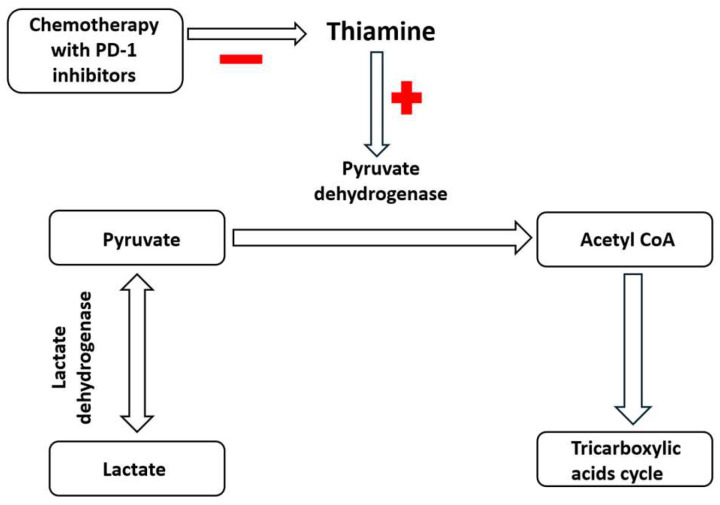
Thiamine stimulates lactate clearance towards the cycle of tricarboxylic acids via pyruvate. Chemotherapy with PD–1 inhibitors has the opposite effect by decreasing thiamine levels. Red minus (−) and red plus (+) indicate inhibition and stimulation respectively.

## Data Availability

The original contributions presented in this study are included in the article. Further inquiries can be directed to the corresponding author.

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
