# Peer review of "Severe Intra- and Post-Operative Lactic Acidosis in a Patient Who Underwent Robotic Thoracoscopic Surgery"

_biomedicines, 2025, doi:10.3390/biomedicines13030568_

Round 1
Reviewer 1 Report
Comments and Suggestions for Authors
This is an interesting case report. This report implies that lactic metabolic acidosis may be induced by chemotherapy and during tumor manipulation by a surgeon, and the successful recovery of blood lactic levels after thiamine treatment may point towards thiamine deficiency as a possible cause of lactic acidosis in this patient, hence is clinically significant.
The manuscript should be formatted through.
Author Response
Reviewer 1
This is an interesting case report. This report implies that lactic metabolic acidosis may be induced by chemotherapy and during tumor manipulation by a surgeon, and the successful recovery of blood lactic levels after thiamine treatment may point towards thiamine deficiency as a possible cause of lactic acidosis in this patient, hence is clinically significant.
The manuscript should be formatted through.
Our response:
We appreciate the reviewer’s positive feedback on the significance of our case report. We have ensured that the ms is formatted according to journal guidelines.
Reviewer 2 Report
Comments and Suggestions for Authors
It would be useful to add the course of arterial blood gases during the operation.
Could you also give your opinion on possible tissue ischemia in the non-ventilated lung (since there was surely single-lung ventilation). Could this explain the increase in lactate?
Author Response
Reviewer 2
It would be useful to add the course of arterial blood gases during the operation.
Our response:
We thank the reviewer for raising this issue and have added the appropriate data on the intraoperative arterial blood gas (ABG) values to Figure 1.
Could you also give your opinion on possible tissue ischemia in the non-ventilated lung (since there was surely single-lung ventilation). Could this explain the increase in lactate?
Our response:
Thank you for this critical question. We added to Discussion the following: “Single-lung ventilation can potentially contribute to transient regional hypoxia and ischemia in the non-ventilated lung, with a subsequent increase in lactate levels. Yet, based on our experience of approximately three hundred thoracic surgeries per year and the corresponding protocols on one-lung ventilation (OLV), such a dramatic increase in lactate levels is hardly probable. The pulmonary vasoconstriction mechanism (PVC) re-duces the lung blood supply to non-ventilated lung, thus partially adjusting O2 delivery to tissue demand. All in all, this commonly does not result in lung ischemia and lactic acid accumulation secondary to PVC.” [lines 168-175]
The appropriate reference is now cited and added to References:
Shum S, Huang A, Slinger P. Hypoxaemia during one lung ventilation. BJA Educ. 2023 Sep;23(9):328-336.
Reviewer 3 Report
Comments and Suggestions for Authors
Dear authors,
Thank you for the opportunity to review your manuscript on a compelling case of severe intraoperative lactic acidosis in a patient undergoing robotic thoracoscopic lobectomy. Your work presents a valuable contribution to understanding perioperative metabolic derangements, particularly in the context of malignancy and potential thiamine deficiency. Below, I provide specific suggestions to enhance the clarity, impact, and scientific rigor of your manuscript.
Abstract - clarify the conclusion: The abstract suggests thiamine deficiency as a possible cause, but the lack of thiamine level measurement weakens this claim. A more balanced phrasing (e.g., "suggestive of" rather than "pointing towards") would improve credibility.
Introduction - provide more background on lactic acidosis in thoracic surgery: While general concepts of lactic acidosis are covered, the introduction should better contextualize its relevance to robotic thoracic surgery.
- clarify research gap: The introduction states that this is the first case report of severe intraoperative lactic acidosis in robotic thoracic surgery. Supporting this with a brief review of existing cases of lactic acidosis in non-robotic thoracic surgery would strengthen the novelty claim.
Case description - clarify perioperative hemodynamic: The patient was described as "hemodynamically stable," yet significant lactate accumulation occurred. A more detailed breakdown of blood pressure and cardiac output trends could provide clarity.
- better describe tumor resection timing: The case suggests that tumor manipulation might have triggered lactate elevation. More precise timing relative to surgical steps would strengthen this hypothesis.
Discussion: article suggests tumor metabolism as a contributor but does not discuss why this is uncommon in similar cases. The discussion also relies on a post hoc response to thiamine administration. If thiamine levels were not measured, discuss why preoperative screening could be beneficial in patients with prior chemotherapy exposure.
Conclusion: given the lack of direct thiamine measurement, avoid making definitive claims about its deficiency. Instead, conclude that "thiamine deficiency should be considered in similar cases." Briefly outline how this case impacts perioperative management, such as possible preoperative thiamine screening or monitoring lactate trends intraoperatively.
Author Response
Reviewer 3
Thank you for the opportunity to review your manuscript on a compelling case of severe intraoperative lactic acidosis in a patient undergoing robotic thoracoscopic lobectomy. Your work presents a valuable contribution to understanding perioperative metabolic derangements, particularly in the context of malignancy and potential thiamine deficiency. Below, I provide specific suggestions to enhance the clarity, impact, and scientific rigor of your manuscript.
Abstract - clarify the conclusion: The abstract suggests thiamine deficiency as a possible cause, but the lack of thiamine level measurement weakens this claim. A more balanced phrasing (e.g., "suggestive of" rather than "pointing towards") would improve credibility.
Our response:
We agree with this suggestion and have made the appropriate changes. Now, instead of “…might point towards thiamine deficiency…”, it is written that our findings are suggestive of thiamine deficiency. [line 26]
Introduction - provide more background on lactic acidosis in thoracic surgery: While general concepts of lactic acidosis are covered, the introduction should better contextualize its relevance to robotic thoracic surgery.
Our response:
We extended the introduction on lactic acidosis in thoracic surgery, as requested: “Several operation types can induce lactic acidosis with transient elevation of lactate, such as surgeries using cardiopulmonary bypass like CABG and aortic surgeries, major hepatectomy, and orthopedic surgery with tourniquet. There are no reports on severe metabolic acidosis in robotic or non-robotic thoracic surgeries.” [lines 52-55]. To the best of our knowledge, there are no publications on severe lactic acidosis as a result of robotic thoracic surgery or the video-assisted thoracic operation.
The appropriate references are now cited and added to References:
Demers P, Elkouri S, Martineau R, Couturier A, Cartier R. Outcome with high blood lactate levels during cardiopulmonary bypass in adult cardiac operation. Ann Thorac Surg. 2000 Dec;70(6):2082-6.
Honore PM, Jacobs R, Hendrickx I, De Waele E, Spapen HD. Lactate: the Black Peter in high-risk gastrointestinal surgery patients. J Thorac Dis. 2016 Jun;8(6):E440-2.
Clarify the research gap: The introduction states that this is the first case report of severe intraoperative lactic acidosis in robotic thoracic surgery. Supporting this with a brief review of existing cases of lactic acidosis in non-robotic thoracic surgery would strengthen the novelty claim.
Our response:
There are no reports of severe metabolic acidosis in robotic or non-robotic thoracic surgeries. [lines 54-55]
Case description - clarify perioperative hemodynamic: The patient was described as "hemodynamically stable," yet significant lactate accumulation occurred. A more detailed breakdown of blood pressure and cardiac output trends could provide clarity.
Our response:
We clarified perioperative hemodynamics: “The patient was hemodynamically stable, with minimal phenylephrine support throughout the perioperative period. Blood pressure variations were within 15-20% of its baseline, and mean arterial pressure was >65 mm Hg. If needed, the transient decrease in the patient's blood pressure was corrected by administrating small boluses of phenylephrine 50-100 mcg. The pulse pressure variation was normal and did not exceed 13% throughout the surgery.” [lines 90-96] Unfortunately, we did not measure cardiac output.
- better describe tumor resection timing: The case suggests that tumor manipulation might have triggered lactate elevation. More precise timing relative to surgical steps would strengthen this hypothesis.
Our response:
We appreciate this suggestion and added the arrows to Figure 1, indicating the precise timing of tumor manipulation and resection.
Discussion: The article suggests tumor metabolism as a contributor but does not discuss why this is uncommon in similar cases. The discussion also relies on a post hoc response to thiamine administration. If thiamine levels were not measured, discuss why preoperative screening could be beneficial in patients with prior chemotherapy exposure.
Our response:
We thank the reviewer for raising these points. Indeed, there are sporadic studies suggesting that, due to Warburg effect, some epithelial tumors could occasionally produce lactic acidosis. In addition, chemotherapy per se could induce thiamine deficiency. We stressed this point, with referring to the relevant references: “In addition, chemotherapy per se could induce subclinical thiamine deficiency (Figure 3). In particular, in one lung cancer study, thiamine deficiency was reported when the patient received Nivolumab [18]. Also, PD-1 inhibitors (Nivolumab, Pembrolizumab) may inhibit pyruvate dehydrogenase in mitochondria, thus promoting lactate accumulation [19]. Altogether, preoperative screening could be helpful in patients with prior chemotherapy exposure to programmed cell death protein 1 inhibitors (PD-1).” [lines 184-189].
Conclusion: given the lack of direct thiamine measurement, avoid making definitive claims about its deficiency. Instead, conclude that "thiamine deficiency should be considered in similar cases." Briefly outline how this case impacts perioperative management, such as possible preoperative thiamine screening or monitoring lactate trends intraoperatively.
Our response:
Though the pathogenesis of this state in our case is not fully clear, it could, to some extent, be induced by chemotherapy and during tumor manipulation by a surgeon. We added the following piece: “Yet, due to the successful recovery of blood lactic levels after thiamine treatment, it could be speculated that thiamine deficiency might be a possible cause of lactic acidosis in our patient. Although we do not have data on the plasma thiamine level, we suggest that its determination in the perioperative period should also be considered in similar cases.” [lines 162-166].
Reviewer 4 Report
Comments and Suggestions for Authors
-
Introduction:
-
Improve context for robotic surgery: The unique risks of robotic thoracoscopic surgery (e.g., prolonged CO₂ insufflation, lateral positioning effects on hepatic perfusion) are underdeveloped. Contrast lactic acidosis rates in robotic vs. non-robotic thoracic procedures.
-
Strengthen chemotherapy relevance: Discuss PD-1 inhibitors (Nivolumab/Pembrolizumab) and their potential mitochondrial toxicity (e.g., pyruvate dehydrogenase inhibition).
-
-
Case Description:
-
Missing data: Report end-tidal CO₂ trends, insufflation pressures, and hemodynamic metrics (e.g., stroke volume variation). Justify restrictive fluid strategy (100 mL/h) against ERAS guidelines.
-
Clarify interventions: Specify adjustments to vasopressors (phenylephrine) during acidosis progression.
-
-
Results:
-
Label figures comprehensively: Add units (e.g., lactate in mmol/L) and intraoperative milestones (e.g., tumor resection) to Figure 1. Redesign Figure 3 to illustrate chemotherapy’s metabolic effects.
-
-
Discussion:
-
Address robotic-specific mechanisms: Dedicate a subsection to robotic surgery contributors (e.g., compartment syndrome from instrument pressure).
-
Temper thiamine claims: Acknowledge lack of plasma thiamine levels and expand on PD-1 inhibitors’ role in lactate metabolism.
-
Strengthen differential diagnosis rigor: Note limitations in excluding mesenteric ischemia (e.g., deferred CT angiography).
-
-
Conclusions:
-
Replace unsupported recommendations (e.g., perioperative thiamine testing) with hypothesis-driven proposals (e.g., “thiamine prophylaxis in chemotherapy-exposed patients warrants investigation”).
-
-
Language:
-
Clarify causal language (e.g., “might point towards” should be something like “suggests association”).
-
The English could be improved to more clearly express the research.
Author Response
Reviewer 4
Introduction:
Improve context for robotic surgery: The unique risks of robotic thoracoscopic surgery (e.g., prolonged CO₂ insufflation, lateral positioning effects on hepatic perfusion) are underdeveloped. Contrast lactic acidosis rates in robotic vs. non-robotic thoracic procedures.
Our response:
We thank the Reviewer for the suggestions regarding robotic thoracoscopic surgery. We find no literature data on comparing lactic acidosis rates between robotic and non-robotic thoracic procedures.
Strengthen chemotherapy relevance: Discuss PD-1 inhibitors (Nivolumab/Pembrolizumab) and their potential mitochondrial toxicity (e.g., pyruvate dehydrogenase inhibition).
Our response:
We added the corresponding sentence with the reference: “Also, PD-1 inhibitors (Nivolumab, Pembrolizumab) may inhibit pyruvate dehydrogenase in mitochondria, thus promoting lactate accumulation [19].” [lines 186-189].
Akbari H, Taghizadeh-Hesary F, Bahadori M. Mitochondria determine response to anti-programmed cell death protein-1 (anti-PD-1) immunotherapy: An evidence-based hypothesis. Mitochondrion. 2022;62:151-158.
Case Description:
Missing data: Report end-tidal CO₂ trends, insufflation pressures, and hemodynamic metrics (e.g., stroke volume variation). Justify restrictive fluid strategy (100 mL/h) against ERAS guidelines.
Our response:
We added info regarding the end-tidal CO₂, insufflation pressure, and fluid restrictive strategy to case description: “…according to ERAS guidelines for thoracic surgery [7]. The end-tidal CO2 varied within the normal range (35-45 mm Hg) in the perioperative period. The ventilation parameters such as tidal volume (TV) and peak inspiratory pressure (PIP) also were accepted for one-lung ventilation protocol (TV 5-6 ml /kg and PIP ≤ 30 mm H2O). The patient was hemodynamically stable, with minimal phenylephrine support throughout the perioperative period. Blood pressure variations were within 15-20% of its baseline, and mean arterial pressure was >65 mm Hg. If needed, the transient decrease in the patient's blood pressure was corrected by administrating small boluses of phenylephrine 50-100 mcg. The total dose was quite small and did not exceed 300 mcg. The pulse pressure variation was normal and did not exceed 13% throughout the surgery.” [lines 86-96]
The appropriate references are now cited and added to References:
Batchelor TJP, Rasburn NJ, Abdelnour-Berchtold E, Brunelli A, Cerfolio RJ, Gonzalez M, Ljungqvist O, Petersen RH, Popescu WM, Slinger PD, Naidu B. Guidelines for enhanced recovery after lung surgery: recommendations of the Enhanced Recovery After Surgery (ERAS®) Society and the European Society of Thoracic Surgeons (ESTS). Eur J Cardiothorac Surg. 2019 Jan 1;55(1):91-115
Clarify interventions: Specify adjustments to vasopressors (phenylephrine) during acidosis progression.
Our response:
If needed, the transient decrease in the patient's blood pressure was corrected by administrating small boluses of phenylephrine 50-100 mcg. The total dose was quite small and did not exceed 300 mcg, which did not require adjustment during acidosis progression. The hemodynamic parameters were stable in the perioperative period. The corresponding piece appears in [lines 92-96]
Results:
Label figures comprehensively: Add units (e.g., lactate in mmol/L) and intraoperative milestones (e.g., tumor resection) to Figure 1. Redesign Figure 3 to illustrate chemotherapy’s metabolic effects.
Our response:
We made the requested changes in Figure 1 and Figure 3, and added explanation to the Figure 3 legend: “Chemotherapy has the opposite effect by decreasing thiamine levels.”
Discussion:
Address robotic-specific mechanisms: Dedicate a subsection to robotic surgery contributors (e.g., compartment syndrome from instrument pressure).
Our response:
The compartment syndrome was reported as an extremely rare (0.028%) complication in robotic-assisted laparoscopic surgery secondary to high intra-abdominal pressure insufflation of more than 20 mm Hg over an extended period*. No similar observations were reported for robotic-assisted thoracoscopic surgery.
*Maerz DA, Beck LN, Sim AJ, Gainsburg DM. Complications of robotic-assisted laparoscopic surgery distant from the surgical site. Br J Anaesth. 2017 Apr 1;118(4):492-503.
Temper thiamine claims: Acknowledge lack of plasma thiamine levels and expand on PD-1 inhibitors’ role in lactate metabolism.
Our response:
We have now discussed PD-1 inhibitors and their potential impact on mitochondrial function, particularly regarding pyruvate dehydrogenase inhibition and its role in lactate metabolism [lines 186-189]. In Conclusions, we emphasized that “we do not have data on the plasma thiamine levels…” [line 196-197]
Strengthen differential diagnosis rigor: Note limitations in excluding mesenteric ischemia (e.g., deferred CT angiography).
Our response:
We addressed this point in the Discussion: “… a mesenteric event was ruled out because of negative clinical findings and improved lactate levels in the ICU, so that CT angiography was not relevant in our case.” [lines 146-149]
Conclusions:
Replace unsupported recommendations (e.g., perioperative thiamine testing) with hypothesis-driven proposals (e.g., “thiamine prophylaxis in chemotherapy-exposed patients warrants investigation”).
Our response:
We made corresponding changes in Conclusions: “Although we do not have data on plasma thiamine levels, we suggest that determining them in the perioperative period could be beneficial for robotic thoracoscopic surgeries. Whatever the case, thiamine prophylaxis in chemotherapy-exposed patients who are planned for robotic thoracoscopic surgeries warrants further investigation.” [lines 196-200]
Language:
Clarify causal language (e.g., “might point towards” should be something like “suggests association”).
Our response
We agree with this suggestion and have made the appropriate changes. Now, instead of “…might point towards thiamine deficiency…”, it is written that our findings "… are suggestive of thiamine deficiency…” [line 26]
Round 2
Reviewer 4 Report
Comments and Suggestions for Authors
Minor Language Edits
- Perform a final proofread to ensure consistency of verb tenses and clarity in causal statements (e.g., changing “could be induced by” to “may have been induced by” to avoid overstating causality).
- Clarify any specialized abbreviations at first mention (e.g., PVC, PD‐1, OLV).
Figure Labelling
- Ensure that each figure axis has units (e.g., mmol/L for lactate, minutes or hours for time) and clearly mark critical intraoperative events (e.g., resection time point).
- Figure 3 is improved; consider adding a brief note to highlight any direct effect(s) of PD‐1 inhibitors vs. standard chemotherapy on thiamine metabolism or lactate production.
Methods/Case Description
- If no CO₂ insufflation was utilized (or if the pressure is negligible in thoracic robotic surgery), briefly note that in the text to show it was considered.
- Reiterate the rationale for the initial restrictive fluid approach, as recommended by ERAS guidelines, even though fluid boluses ultimately became necessary.
Conclusions
- If possible, add a short note that the Warburg effect is speculative in this case, given limited direct tumor metabolism measurements.
Comments on the Quality of English Language
Overall readability is improved, though a final minor proofreading pass is advisable for example ensure consistent tense usage, clarity of causality/association statements
Author Response
Minor Language Edits
Perform a final proofread to ensure consistency of verb tenses and clarity in causal statements (e.g., changing “could be induced by” to “may have been induced by” to avoid overstating causality).
Our response:
We made the requested changes. Now, instead of “…could be induced by …”, it is written “…may have been induced by…” [lines 25,198]. We also carefully reviewed the ms and corrected typos, inconsistencies, etc.
Clarify any specialized abbreviations at first mention (e.g., PVC, PD‐1, OLV).
Our response:
We clarified specialized abbreviations at first mention (PVC, PD‐1, OLV). [lines 174,175,188].
Figure Labelling
Ensure that each figure axis has units (e.g., mmol/L for lactate, minutes or hours for time) and clearly mark critical intraoperative events (e.g., resection time point).
Our response:
We added the units of the presented parameters to Figure 1, and marked the precise time point of tumor resection.
Figure 3 is improved; consider adding a brief note to highlight any direct effect(s) of PD‐1 inhibitors vs. standard chemotherapy on thiamine metabolism or lactate production.
Our response:
We modified Figure 3 by including in Chemotherapy box "…by PD-1 inhibitors" and added the explanation to the Figure 3 legend: “Chemotherapy with PD-1 inhibitors has the opposite effect by decreasing thiamine levels.” [line 143]. Additionally, we added the discussion of the effects of PD‐1 inhibitors vs. standard chemotherapy on thiamine metabolism and lactate production: “In addition, standard chemotherapy per se could induce a metabolic shift towards anaerobic glycolysis. The programmed cell death protein 1 (PD-1) inhibitors (Nivolumab, Pembrolizumab) may promote lactate accumulation (like the standard chemotherapy) by inhibiting pyruvate dehydrogenase in mitochondria [18]. Also, the PD-1 inhibitors could induce subclinical thiamine deficiency with a subsequent increase in lactate levels (Figure 3). In particular, in one lung cancer study, thiamine deficiency was reported when the patient received Nivolumab [19]. Altogether, preoperative screening could be helpful in patients with prior chemotherapy exposure to PD-1 inhibitors.” [lines 187-195].
Methods/Case Description
If no CO₂ insufflation was utilized (or if the pressure is negligible in thoracic robotic surgery), briefly note that in the text to show it was considered.
Our response:
We addressed this point in the Case Description: “In the intraoperative period, the positive pressure of 5–8 mmHg in the thoracic cavity was established by insufflation of Carbon dioxide (CO2).” [lines 130-132].
Reiterate the rationale for the initial restrictive fluid approach, as recommended by ERAS guidelines, even though fluid boluses ultimately became necessary.
Our response:
We extended the rationale for the initial restrictive fluid approach, as recommended by ERAS guidelines [lines 86-87].
Conclusions
If possible, add a short note that the Warburg effect is speculative in this case, given limited direct tumor metabolism measurements.
Our response:
We added the corresponding sentence to Discussion: “It is attractive to speculate that lactic acidosis may be associated with the so-called "Warburg effect." [lines 182-183].